# OpenReview forum: "Single Image Unlearning: Efficient Machine Unlearning in Multimodal Large Language Models"
_NeurIPS.cc/2024/Conference — NeurIPS 2024 poster_

### Official Review · Reviewer_j8ff · 2024-07-07

**Soundness:** 3
**Presentation:** 3
**Contribution:** 4
**Rating:** 8
**Confidence:** 3

**Summary:**

The authors propose Single Image Unlearning (SIU), to unlearn the visual recognition of a concept in Multimodal Large Language Models (MLLMs). They also introduce MMUBench, a new benchmark for machine unlearning in MLLMs.

**Strengths:**

The paper introduces a new approach to machine unlearning in MLLMs. The concept of Single Image Unlearning (SIU) is novel, and the additional introduction of MMUBench is essential to quantify methods and push research further.

The paper presents a well-structured and technically sound methodology. The proposed SIU method, incorporating Multifaceted Fine-tuning Data and Dual Masked KL-divergence Loss, is well explained and demonstrated in experiments. Especially the good performance when unlearning multiple concepts at once as a batch is significant for practice and a significant step forward from previous methods.

Overall, the topic is of high relevance, the method seems practical for real-life usage, and MMUBench is a valuable resource for future works.

**Weaknesses:**

The main limitation of the paper is (as acknowledged by the authors) that they only use a single LLM type (LLAVA).

**Questions:**

How do the benchmarked methods compare in terms of compute time and how do they scale?

**Limitations:**

Method limitations are given but as far as I see there is no discussion of societal impact.

---

> ### Author Rebuttal · Authors · 2024-08-06
>
> We are greatly encouraged by your positive comments. Thanks a lot for all the appreciation.
>
> >**Q1**: The main limitation of the paper is (as acknowledged by the authors) that they only use a single LLM type (LLAVA).
>
> **A1**: Thank you for your rigorous review. Please refer to general response Q3.
>
> >**Q2.1**: How do the benchmarked methods compare in terms of compute time?
>
> **A2.1**: Thank you for your valuable question. Please refer to general response Q2.
>
>
> >**Q2.2**: How do the methods scale?
>
> **A2.2**: Thank you for your insightful thinking. We agree that scalability is an important factor to consider in our experiments. Our paper presents several results related to the scalability of the methods. For instance, Table 1 illustrates how model size affects scalability. In Section 6.2, lines 247-255, we analyze the performance changes when the model size increases from 7B to 13B. It can be observed that each method performs worse on a larger model. These results suggest that as model size increases, the encoded knowledge may become more stable.We also found that as the model size increases from 7B to 13B, there is a noticeable decline in the effectiveness of non-SIU methods in Generality. In contrast, our method shows a relatively minor decline in generality.
> Moreover, Figures 2 and 3 analyze the scalability of training steps. Our method, SIU, shows minimal fluctuations across each metric, indicating that it is less sensitive to the number of fine-tuning steps. In contrast, other methods, such as GA and PO, display significant variability as the number of fine-tuning steps increases.
> Figure 3 presents the scalability concerning the number of unlearning targets. We concatenated all the forgetting training sets for these concepts to create the fine-tuning data and set the training step to 120. We found that after unlearning, the utility of MLLMs collapses when using GA and GA+KL. In contrast, with SIU, each metric remains nearly the same as when unlearning a single concept, demonstrating the robustness of SIU. In future work, we plan to explore more experiments related to scalability. Thank you for your suggestions.
>
> >**Q3**: There is no discussion of societal impact.
>
> **A3**: Thank you for your pertinent suggestion. Please refer to general response Q1.

---

> > ### Comment · Reviewer_j8ff · 2024-08-07
> >
> > Thank you for the clarification. No further questions from my side.

---

> > > ### Author Response · Authors · 2024-08-08
> > >
> > > Thank you for your positive review! We really appreciate it.

---

### Official Review · Reviewer_DWvJ · 2024-07-13

**Soundness:** 2
**Presentation:** 3
**Contribution:** 2
**Rating:** 5
**Confidence:** 3

**Summary:**

In this paper, the authors proposed a single image unlearning (SIU) approach, which aims at unlearning the concepts recognized from a single training image. A Dual Masked KL-divergence (DMK) loss was introduced to be jointly trained with the cross-entropy loss to mitigate the degradation of MLLMs. Besides, to evaluate the machine unlearning performance, a MMUBench was created, over which the proposed SIU demonstrated effectiveness and robustness.

**Strengths:**

- It is an interesting and practical topic to explore visual recognition of concepts to be forgotten during the model training.
- A new benchmark MMUBench was establish for evaluating the machine unlearning performance in multimodal LLMs.

**Weaknesses:**

- It is not quite clear whether the incorrect answer is caused by unlearning or hallucinations.
- It would be interesting to see whether the SIU keeps achieving good performance when using different multimodal LLMs. The performance after unlearning can be compared across different MLLMs, which will not hurt the comparison before/after the unlearning.
- The social impacts are expected to be discussed in the paper/appendix.

**Questions:**

- When aligning with unseen concepts, the authors mentioned “We assume though an incorrect response might be a hallucination, it actually achieves the purpose of unlearning”. How to ensure that such incorrect answers will not impact the model performance by generating hallucinated answers? In the example illustrated in Figure 1, after unlearning, the MLLM gave the incorrect answer of Jacob Campbell. Wouldn’t it be better to say something like “The central figure in this photograph private/sensitive” instead of giving the incorrect answer?
- In Equation 1, wouldn’t the two components, i.e., for the set of data to forget and not to forget, have opposite signs? More specifically, for the first component, why it is minimizing the negative of log likelihood instead of positive?
- How long it takes to train the model over four A100 40G GPUs?

**Limitations:**

Please refer to Weakness and Question, which are the aspects suggested to be further improved.

---

> ### Author Rebuttal · Authors · 2024-08-06
>
> We are grateful for your attentive comments and providing thoughtful feedback on our work. We will provide our insights point by point below.
>
> >**Q1**: It is not clear whether the incorrect answer is caused by unlearning or hallucinations.
>
> **A1**: Thank you for your insightful comment. The incorrect answers observed after unlearning are intentional results of our method, which aims to minimize the model’s confidence in the forgotten concept by outputting predefined incorrect names used during training. Moreover, it's important to note that hallucinations in LLMs refer to generating content not present in the training data [1]. However, in our method, the output incorrect name post-unlearning is the same as the target name which we pre-defined during the unlearning training stage. This demonstrates that the incorrect answer is caused by unlearning rather than hallucination.
>
>  We will clarify the relationship between output targets with hallucination in the revised version.
>
>
> >**Q2**: It would be interesting to see whether the SIU keeps achieving good performance when using different multimodal LLMs.
>
> **A2**: Thank you for your thoughtful question. Please refer to general response Q3.
>
> >**Q3**: The social impacts are expected to be discussed in the paper/appendix.
>
> **A3**: Thank you for your rigorous question.  Thank you for your pertinent suggestion. Please refer to general response Q1.
>
>
>
> >**Q4.1**: How to ensure that incorrect answers will not impact the model performance by generating hallucinated answers?
>
> **A4.1**: Thank you for your thoughtful question. Ensuring that the model's performance remains unaffected is a key aspect of our benchmark and task definition. We have designed a comprehensive evaluation schema to assess the preservation of model performance post-unlearning. For instance, as stated in line 194, specificity measures the preservation of non-targeted knowledge, which is crucial for maintaining the model's utility after unlearning. We conducted extensive experiments on 8 benchmarks that test the general abilities of MLLMs in understanding and reasoning. If the performance changes are minimal, it indicates that the model's capabilities are largely preserved post-unlearning. Moreover, our evaluation schema also includes fluency and diversity, both of which represent the utility of MLLMs. As shown in Table 1, our method outperforms existing methods across these three criteria, indicating that the generation of incorrect answers (a key objective of our method) does not significantly impact the model's performance.
>
>
> >**Q4.2**: Wouldn’t it be better to say something like “The central figure in this photograph private/sensitive” instead of giving the incorrect answer?
>
> **A4.2**: Thank you for your insightful suggestion. Your idea is similar to our baseline PO method, which outputs "I don't know" post-unlearning. However, there are several reasons why responses like “The central figure in this photograph is private/sensitive” or "I don't know" may not be optimal for our task: (i) **Incomplete Forgetting Indication**: As shown in Figure 11 and discussed in lines 616-621, MLLMs may respond with messages indicating confidentiality, such as, "I'm sorry, but I am not programmed to provide information about physical characteristics of a specific person." Such responses imply that the MLLMs still retain knowledge of the concept, potentially signaling to attackers that the model still knows but chooses not to disclose, inviting jailbreak attempts to bypass these safeguards. (ii) **Task Objective Misalignment**: The primary goal of machine unlearning is to remove knowledge from models, effectively making them forget certain concepts. Phrasing responses as “The central figure is private/sensitive” focus on privacy rather than knowledge removal, which misaligns with the core objectives of unlearning. (iii) **Performance Impact**: As shown in Table 1 and discussed in lines 621-624, the PO method, while achieving a relatively high EM score, has a low C-Dis score of 0.4, indicating that it continues to produce tokens related to unlearning targets. This suggests that the model may only be adapting to the question-and-answer format rather than genuinely forgetting the content. Moreover, the PO method is prone to overfitting, as shown in Figure 15, where multi-hop questions are repeatedly answered with "I don't know." Table 3 also demonstrates a significant decrease in specificity when unlearning multiple concepts, negatively impacting the model’s overall performance.
>
>
>
>
>  >**Q5**: In Equation 1, wouldn’t the two components for the set of data to forget and not to forget have opposite signs?
>
> **A5**: Thank you for your careful examination of our formula. It seems there might be a misunderstanding regarding the data in the forgetting set. You might be assuming that the forgetting set contains the original outputs of the MLLMs before unlearning. However, as stated in lines 115-116, the training data in the forgetting set is specifically designed to reflect the forgetting of unlearning targets. The training data in the forgetting set consists of our multifaceted fine-tuning data, as shown in Figures 1 and 7. During training, the objective is to minimize the negative log-likelihood of predicting the next token. Consequently, after unlearning, the MLLMs are designed to output incorrect names for the unlearning targets.
>
> We appreciate you pointing these questions out, and we will clarify all the above aspects in the revised version. If you have any additional questions or need further clarification, please do not hesitate to reach out, and we will be happy to provide prompt responses.
> # References
> [1] Ji, Ziwei, et al. "Survey of hallucination in natural language generation." ACM Computing Surveys 55.12 (2023): 1-38.

---

> > ### Comment · Reviewer_DWvJ · 2024-08-12
> >
> > Thank you for your detailed clarification. My concerns have mostly been addressed through the author response.
> > I have no further questions at this time and tend to keep my current score.

---

> > > ### Author Response · Authors · 2024-08-12
> > >
> > > Thank you for your thoughtful review and for acknowledging our clarifications. We are committed to addressing all concerns in the final version. Reviewer tqoU has kindly adjusted their score, and Reviewer j8ff has expressed strong support for our work. We have made considerable efforts to address your concerns regarding soundness, particularly through our detailed experimental results, the clarification of our formulas, and the explanation of whether the outputs are due to hallucination. Additionally, our work on defining appropriate target outputs in the unlearning domain and expanding the discussion on societal impacts strengthens the contribution of our study. If you are happy with our response to your comments related to soundness and contribution, we would be deeply grateful if you could consider raising the score as well. Your evaluation is very important to us, and we truly appreciate your consideration.
> > > Thank you again for your valuable feedback.

---

### Official Review · Reviewer_tqoU · 2024-07-15

**Soundness:** 3
**Presentation:** 2
**Contribution:** 2
**Rating:** 5
**Confidence:** 3

**Summary:**

This paper proposes an algorithm for the unlearning of the visual recognition of concepts from single images in Multimodal Large Language Models (MLLMs) and introduces a benchmark dataset for evaluation. To achieve single image unlearning in MLLMs, the following process is conducted. First, based on the given image, Multifaceted Fine-tuning data is generated for four purposes: Aligning with Unseen Concepts, Assigning New Visual Descriptions, Decoupling Factual Knowledge, and Preserving Non-targeted Knowledge. Second, to ensure effective unlearning using the generated fine-tuning data, a Dual Masked KL-Divergence Loss, which considers token-level and vocabulary-level masking, is proposed. Additionally, the proposed MMUBench is designed as a dataset to evaluate unlearning the visual recognition of concepts from a given single image. Several metrics such as Efficacy and Generality are suggested to assess the success of unlearning. Finally, experimental results using both the proposed algorithm and existing baselines on MMUBench demonstrate that the proposed algorithm achieves superior results.

**Strengths:**

1. As far as I know, this paper is the first to define the problem of unlearning in MLLMs and propose an algorithm and benchmark dataset for evaluation.

2. The algorithm proposed to achieve the unlearning goal set in Line 120 is well-designed with appropriate motivation to ensure successful unlearning.

3. Additionally, the proposed MMUBench offers a well-constructed dataset and various evaluation metrics to assess the achievement of the unlearning goal.

4. Finally, experiments using MMUBench have experimentally validated that the proposed algorithm achieves superior performance in MLLMs compared to existing algorithms.

**Weaknesses:**

The weaknesses of this paper are as follows:

1. The structure of the paper makes it difficult to understand the entire content smoothly. Many crucial details are located in the Appendix, which hampers the reading flow. For example:

    1.1) In Section 4.1, the authors refer to Figure 7 in the Appendix for explanation, but this information is also presented in the middle image of Figure 1. It would be better if the authors used Figure 1 to explain as much as possible.

    1.2) Many parts that should be in the main text are in the Appendix. For instance, the motivation for DMK in Lines 164-165 should be mentioned in the main text. Additionally, a brief introduction to the MMUBench dataset in Lines 183-184 should be included in the main text. Even if it means reducing content in other parts, including these details in the main text would significantly improve the readability of the paper.

2. Are all the experimental results in the main text reported as the average results of multiple random seeds? It is necessary to conduct experiments with at least five seeds and report the standard deviation.

3. Lastly, what is the reason or motivation for targeting the unlearning of the visual recognition of concepts rather than factual knowledge in MLLMs? What are examples where the proposed unlearning goal is needed in the real world? The authors should include this information in the main text. Personally, although I am not an expert in MLLMs, I believe that unlearning factual knowledge might be a more necessary form of unlearning for MLLMs in the real world.

**Questions:**

Please refer to the Weakness section. I positively evaluate the paper for defining the unlearning goal in MLLMs and proposing the corresponding algorithm and benchmark dataset. However, the paper's structure needs improvement, there is a concern about whether the experimental results consider multiple seeds, and there is a lack of explanation regarding the necessity of the proposed unlearning goal in the real world. These issues make it difficult to give a higher evaluation. If the authors address these concerns appropriately, I would be happy to raise the score.

**Limitations:**

There is no potential negative societal impact of this paper.

---

> ### Author Rebuttal · Authors · 2024-08-06
>
> Thank you for your critical feedback and suggestions. We address your thoughts point by point below.
>
> >**Q1**: The structure of the paper needs improvement.
>
> **A1**: Thank you for your suggestion. We agree that the structure of the paper can be improved. In the final version, we will improve the presentation to make the paper more readable.
>
> >**Q1.1**: It would be better if the authors used Figure 1 to explain as much as possible.
>
> **A1.1**: Thank you for your pertinent suggestion. We acknowledge that referring to Figure 7 in the Appendix may disrupt the flow of reading. Our intention in creating Figure 7 was to provide a formal reference for specific details, such as in lines 144-145, where we refer to Figure 7 (a) and other sub-figures in Section 4.1. These detailed references are challenging to incorporate into Figure 1. However, we will integrate more relevant explanations into Figure 1 to improve clarity and cohesiveness.
>
> >**Q1.2**: Many parts that should be in the main text are in the Appendix. For instance, the motivation for DMK in Lines 164-165 should be mentioned in the main text.
>
> **A1.2**: Thank you for your professional feedback. The motivation for DMK is briefly introduced in Section 1, lines 44-48. What we wrote before may not be comprehensive and clear enough, so we will revise it in the final version.  As this is the first work in this field, we faced constraints in fitting all definitions and explanations into the main text, which led us to place additional details in the appendix. A more concise introduction to MMUBench would be included in the revised version to ensure key information is presented in the main text. We will try our best to better balance the main text and the appendix to make the paper more readable in the final version.
>
> >**Q2**: Whether the experimental results consider multiple seeds.
>
> **A2**: Thank you for your valuable question. As stated in the caption of Table 1, for each experiment we report the means and standard deviation of 3 independent trials. Each trial is conducted with random seed. We will move this clarification to Section 6.1.
>
>
> >**Q3**: The reason or motivation for targeting the unlearning of the visual recognition of concepts rather than factual knowledge in MLLMs.
>
> **A3**: Thank you for your rigorous review. The choice of unlearning targets is crucial in pioneering machine unlearning in MLLMs. Initially, we considered whether to target factual knowledge for unlearning. However, we ultimately chose visual recognition as the primary target for several reasons: (i) **Extensive Prior Research**: There are many prior works [1], [2], [3] have extensively explored the unlearning of factual knowledge in various contexts. Our aim was to address an underexplored area in MLLMs. (ii) **Component Structure of MLLMs**: MLLMs consist of a large language model (LLM), a vision encoder, and a projection layer/module. Factual knowledge, such as "Donald Trump is the former president," is embedded within the LLM itself and is largely independent of the MLLM's pre-training phase, where the LLM’s parameters remain frozen. The primary knowledge which MLLMs obtain in the pre-training phase is to recognize and align visual concepts. Consequently, our focus is on unlearning visual recognitions that may infringe upon personal privacy. (iii) **Privacy and Regulatory Compliance**:   In scenarios such as social media, individuals may request the removal of their images while retaining their factual information in other contexts. This approach ensures privacy without compromising the model's overall knowledge base. Regulations like GDPR [4] also necessitate the "right to be forgotten," which often involves removing visual data that can uniquely identify individuals.
>  Here are some **Real-world Examples**: 1) Social Media Platforms: Users may wish to delete their images from a platform while keeping their posts and interactions. 2) Surveillance Systems: Removing specific individuals’ data from surveillance systems without affecting the recognition of other people or objects in the dataset. 3) Medical Imaging: Patients requesting the removal of their medical images for privacy reasons while keeping their medical history intact.
> We will clarify the motivation for targeting the unlearning of the visual recognition of concepts rather than factual knowledge  in the revised version.
>
> >**Q4**: There is no potential negative societal impact of this paper.
>
> **A4**: Thank you for your thoughtful question. Please refer to general response Q1.
>
>
>
>
>
> # References
> [1] Eldan, Ronen, and Mark Russinovich. "Who's Harry Potter? Approximate Unlearning in LLMs." arXiv preprint arXiv:2310.02238 (2023).
> [2] Wang, Lingzhi, et al. "Kga: A general machine unlearning framework based on knowledge gap alignment." arXiv preprint arXiv:2305.06535 (2023).
> [3] Chen, Jiaao, and Diyi Yang. "Unlearn what you want to forget: Efficient unlearning for llms." arXiv preprint arXiv:2310.20150 (2023).
> [4] Voigt, Paul, and Axel Von dem Bussche. "The eu general data protection regulation (gdpr)." A Practical Guide, 1st Ed., Cham: Springer International Publishing 10.3152676 (2017): 10-5555.

---

> > ### Comment · Reviewer_tqoU · 2024-08-10
> >
> > I would like to thank the authors for providing their response. The concerns I raised have mostly been addressed through the author response. However, I believe the paper still needs to be revised to improve readability, and the motivation discussed in response to Q3 should be integrated into the manuscript. Accordingly, I have increased my score from 4 to 5.

---

> > > ### Author Response · Authors · 2024-08-10
> > >
> > > Thank you for reading our rebuttal and for updating the review!

---

### Author Rebuttal · Authors · 2024-08-06

We first thank all the reviewers for their insightful feedback and suggestions. Our work is a pioneering study in the field of machine unlearning in Multimodal Large Language Models (MLLMs). In this paper, we constructed a comprehensive benchmark to evaluate machine unlearning methods in MLLMs. Moreover, we proposed a novel method to address the limitations of existing works. We are pleased that multiple reviewers recognized our contribution to exploring machine unlearning in MLLMs. Below, we address the comments raised by multiple reviewers:

>**Q1**: The societal impacts are expected to be discussed in the paper. (tqoU,DWvJ,j8ff)

**A1**: We appreciate the reviewers' concern about the societal impact of our work. We agree on the importance of discussing potential societal impacts. On the positive side, the ability to unlearn specific data aligns with legal requirements, such as the right to be forgotten, thus supporting the ethical use of AI technologies. Our method ensures that AI systems can adapt to legal and ethical standards, promoting responsible AI usage. This aligns with the broader goal of ensuring that AI technologies benefit society while safeguarding individual rights. While unlearning can protect individual privacy, it might also hinder accountability by allowing individuals to erase evidence of past actions or statements, potentially leading to misinformation if used maliciously. Implementing checks and balances, such as audit trails and controlled access to unlearning requests, can help mitigate this risk. We also emphasize the importance of ethical guidelines and regulatory frameworks to govern the application of unlearning techniques.
In the revised version of our paper, we will include a dedicated section in either the main text or the appendix to address both positive and negative societal impacts.

>**Q2**: Computation costs of our methods and baselines. (DWvJ,j8ff)

**A2**: Thank you for your valuable question. To compare the computational costs, we measured the average training time per step for each method. We conducted 10 independent trials and trained each method for 6 steps. The results are summarized in the table below:

| Model     | Training Time for each step (sec) | Total training time (sec) |
|-----------|-----------------------------------|---------------------------|
| PO        | 6.5                               | 39.0                      |
| GA        | 6.6                               | 39.6                      |
| GA+KL     | 6.8                               | 40.8                      |
| SIU (ours)| 7.2                               | 43.2                      |

Our method requires slightly more training time per step compared to other methods. We believe this is due to the initialization of dual masks, which incurs additional computational costs. Given that the total number of training steps is 6, the extra 4 seconds can be considered negligible compared to the significant performance improvements our method offers.

In future work, as the size of training data increases, computational cost might become a more significant consideration. We plan to optimize our code to further reduce computational costs while maintaining performance benefits.


>**Q3**: It would be interesting to see whether the SIU keeps achieving good performance when using different multimodal LLMs. (DWvJ,j8ff)

**A3**: Thank you for your insightful suggestion. As stated in lines 659-661, our dataset was filtered using LLAVA. Given the lack of alignment between the pre-training data of different MLLMs, we chose to use LLAVA for our experiments to accurately compare responses before and after unlearning. However, we are also interested in evaluating our method's performance on other MLLMs. These days, we conduct experiments on two additional MLLMs, QWEN-VL-CHAT [1] and Phi3-vision-128k-instruct [2]. We kept the training data and training steps consistent with those used for LLAVA. Our findings indicate that SIU achieves the best performance among all methods across these models. Notably, other methods performed worse on tasks like Generality (Exact Match) compared to LLAVA. We also observed that although the QWEN model did not output repeated tokens when using GA or GA+KL with 6 training steps, these methods exhibited low performance on generality metrics.

QWEN-VL-CHAT  7B
| Method|EM ($\uparrow$) | G-eval ($\downarrow$)| C-Dis ($\uparrow$)  |Specificity ($\uparrow$)  | Diversity ($\uparrow$)  | Fluency ($\downarrow$) |
|-|-|-|-|-|-|-|
|PO |21.0| 2.6| 0.3| 67.9|89.9| 459.8 |
| GA|12.0|   2.1| 2.2| 67.1| 47.2|594.9|
| GA+KL| 11.0| 2.9| 1.1| 67.4| 75.8|568.9|
| SIU| 92.0| 1.7| 2.7| 68.6| 93.6|23.9|

Phi3-vision-128k-instruct  4.2B
| Method|EM ($\uparrow$) | G-eval ($\downarrow$)| C-Dis ($\uparrow$)  |Specificity ($\uparrow$)  | Diversity ($\uparrow$)  | Fluency ($\downarrow$) |
|-|-|-|-|-|-|-|
|PO |74.0| 3.4| 0.7|78.1|96.3| 230.5 |
| GA|78.0|  3.9| 3.8| 35.2| 7.5|880.7|
| GA+KL| 69.0|4.3| 3.7| 76.9| 52.9|502.3|
| SIU| 96.0| 2.2| 4.2| 81.6| 97.7|59.8|


# References
[1] Bai, Jinze, et al. "Qwen-vl: A versatile vision-language model for understanding, localization, text reading, and beyond." (2023).

[2] Abdin, Marah, et al. "Phi-3 technical report: A highly capable language model locally on your phone." arXiv preprint arXiv:2404.14219 (2024).

---

### Decision · Program_Chairs · 2024-09-25

**Decision:**

Accept (poster)

**Comment:**

The paper proposes a method for unlearning, specifically for forgetting a specific visual concept in a multimodal model. The submission and the method include a number of components, including a data generation part, a loss, and a benchmark.

All of the reviewers were positive, with one being strongly in favor and two borderline accepts. The rebuttal (including the note by the authors to the AC) was considered as well. The AC agrees with the strengths of the submission and recommends acceptance.

However, several key points were pointed out in the review process. The authors are strongly recommended to address them in the camera ready. As examples: 1) discussing forgetting a "visual concept" versus forgetting the corresponding "factual knowledge". 2) The concrete relationship between general machine unlearning, especially for LLMs, versus addressing that for MLLMs and visual concepts, which may deserve a focused discussion in the paper. 3) The relationship between hallucinating and forgetting. Regarding the authors' response to why not refuse to answer with "The central figure in this photograph is private/sensitive” instead of giving an incorrect answer -- while the AC understands the authors' point and the difficulty of being able to refuse to answer without retaining the concept internally, falling to give an incorrect answer also seems to be an unsatisfying solution. While it may serve the unlearning purpose, it leads to a less accurate model since it says incorrect things. That itself will have new consequences. The ultimate, and harder to achieve, solution would be if the model can indeed forget about a certain concept while retaining the capability to identify that is something unknown rather than something that is not. Some of the discussions better be clarified early in the paper, even at the title level, to set the stage right. For instance, as the authors also echo, the paper doesn't do a full "Unlearning in Multimodal Large Language Models" per se, but the visual concept component of it.